# Preferences for cesarean section among pregnant women at a tertiary hospital in Ho Chi Minh City, Vietnam: Influencing factors and implications for prenatal care

Trang Thi Thuy Hoang[1], Oanh Thi Hoang Trinh[2], Chau Giang Huynh[3*], Truc Thanh Thai[4*], Quan Minh Pham[5], Hue Thi Hoang[6], Dan Trieu Thanh Nguyen[7], Minh Thien Nguyen[2], Hang Thi Phan[7]

**1** Department of General Planning, Hung Vuong Hospital, Ho Chi Minh City, Vietnam, **2** Department of Epidemiology, University of Medicine and Pharmacy at Ho Chi Minh City, Ho Chi Minh City, Vietnam, **3** Department of Gynecologic Oncology and Pathology Department, Hung Vuong Hospital, Ho Chi Minh City, Vietnam, **4** Department of Medical Statistics and Informatics, University of Medicine and Pharmacy at Ho Chi Minh City, Ho Chi Minh City, Vietnam, **5** Department of Fundamental Sciences and Basic Medical, Pham Ngoc Thach University of Medicine, Ho Chi Minh City, Vietnam, **6** Department of Delivery, Hung Vuong Hospital, Ho Chi Minh City, Vietnam, **7** Department of Quality Management, Hung Vuong Hospital, Ho Chi Minh City, Vietnam

* thaithanhtruc@ump.edu.vn (TTT); hgchau.ncs18@ump.edu.vn (CGH)

## Abstract

The World Health Organization recommends a cesarean section (CS) rate of 10–15%. However, global rates may increase to 29% by 2030, raising concerns about the potential overuse of CS without medical indications and its consequences, and in Vietnam, the CS rate reached 34.4% in 2021. This study aims to evaluate Vietnamese women's preferences regarding birth modes and analyze influencing factors, particularly non-medical aspects. This study was conducted at Hung Vuong Hospital in Ho Chi Minh City, Vietnam to survey 599 pregnant women over 36 weeks of gestation. Data were collected through questionnaires covering sociodemographic information, obstetric factors, birth experiences, and knowledge of birth modes. Approximately 27.1% of participants preferred CS. Factors associated with a higher preference for CS included multiparous women with previous CS (adjusted Odds Ratio (aOR) = 41.48, 95% Confidence Interval (CI) 17.56–97.99), fear of complication from vaginal birth (aOR = 6.27, 95% CI: 2.24–17.57), safer for the baby (aOR = 5.08, 95% CI: 2.73–9.46), date of birth affect to the family's life (aOR = 3.48, 95% CI: 1.49–8.13), advice from relatives (aOR = 6.58, 95% CI: 3.24–13.37), recommendation of healthcare provider for CS (aOR = 7.15, 95% CI: 2.49–20.48), exposing negative experience of other women (aOR = 2.66; 95% CI: 1.43–4.98, concern about postpartum sexual activities (aOR = 1.93; 95% CI: 1.04–3.56). In contrast, knowledge of the benefits and drawbacks of mode of birth (aOR = 0.47; 95% CI: 0.26–0.86) and labor companionship expectations (aOR = 0.45; 95% CI: 0.24–0.85) were protective

**Data availability statement:** The data underlying the results presented in the study are available in the appendix (Supporting documents).

**Funding:** Hung Vuong Hospital funded this study (Decision No. 7410/QĐ-BVHV, dated December 15th, 2023). The funders had no role in study design, data collection and analysis, decision to publish, or preparation of the manuscript.

**Competing interests:** The authors have declared that no competing interests exist.

factors, associated with a lower preference for CS. Findings from this study emphasize the need for enhanced prenatal education and counseling in Vietnam to support informed decision-making concerning childbirth, while also emphasizing that clinicians' recommendations are a powerful driver of women's preference for cesarean section, highlighting the importance of targeted strategies in clinical practice.

## Introduction

The method of childbirth is a critical decision for expecting mothers, with significant implications for both maternal and neonatal health. Cesarean section (CS), a surgical procedure where a baby is delivered through incisions in the mother's abdomen and uterus, is often performed when vaginal birth poses risks to the mother or baby's well-being. While CS can be life-saving in emergencies, its overuse without clear medical indications raises concerns about unnecessary risks for mothers and infants [1,2,3]. These risks, which include increased chances of infection, more extended recovery periods, various subsequent pregnancy concerns, including placenta previa, uterine rupture, stillbirth, recurrent CS and potential complications, such as respiratory issues in the infant, underscore the importance of carefully considering when and how cesarean sections should be used in clinical practice, ensuring they are applied based on medical necessity rather than non-medical factors [1,4]. For this reason, the World Health Organization recommends that the CS rate ideally fall between 10% and 15% [4,5].

However, there is a significant disparity in the rate of CS across regions and countries. For instance, in the United States, the CS rate is notably high, with approximately 32% of all births being cesarean deliveries.[6] Similarly, the CS rate in England is also high, reaching around 31% in 2023 [7]. This rate is relatively lower in Japan, at about 20%, but has gradually increased [8]. In contrast, in Africa, some regions have lower CS rates due to limited access to surgical facilities, while other areas report rates closer to the global average, reflecting disparities in healthcare access and quality across the continent [9]. In Thailand, the CS rate has similarly been increasing, with recent statistics showing an approximately 45% rate [10].

In Vietnam, the CS rate has increased significantly over the past few decades, rising from 3.4% in 1994 to 20% in 2011, to 27.5% in 2014, and to 34.4% in 2021 [9,11]. Compared to the national CS rate of approximately 27.5% reported in UNICEF [11], some regions and hospitals reported even higher rates. For example, a recent study in Da Nang revealed a CS rate of 58.6%, with public hospitals at 57.9% and private hospitals at 70.6% in 2016 [9,12]. Additionally, the CS rate at Hung Vương Hospital was approximately 47.6% (2,333 CS out of 4,900 deliveries) in 2017 [13]. These figures far exceed the World Health Organization's recommended threshold, highlighting the potential overuse of CS in the country. The absence of practical monitoring and control strategies may explain the high frequency of CS in Vietnam. This underscores the urgent need for better monitoring and control strategies to ensure the appropriate use of CS. Moreover, the rising incidence of CS could be attributed

to social concerns related to culture. For example, in Vietnamese culture, especially in isolated areas, a person's destiny is partly determined by the day and hour of their birth, and thus, people may urge for CS so that the baby is born at the pre-defined time.

Despite the increasing prevalence of CS, there is a significant gap in our understanding of pregnant women's perspectives regarding birth modes and the factors influencing these choices. This is possibly due to the lack of research on this topic in Vietnam. As healthcare providers, we must bridge this gap and gain a deeper understanding of these preferences and the non-medical aspects that affect them. This understanding is beneficial and essential in balancing the benefits of necessary interventions with the drawbacks of unwarranted cesarean deliveries. Therefore, our study aims to assess the preferences of women receiving care at Hung Vuong Hospital, a major tertiary referral maternity hospital in Ho Chi Minh City, regarding birth modes and analyze the influencing factors, particularly for non-medical purposes. By gaining insights into these preferences, healthcare practitioners can better support expectant mothers in making informed decisions about their childbirth options.

## Materials and methods

### Setting and participants

This study was conducted from 19th April 2023–04th June 2023 at Hung Vuong Hospital, a public tertiary maternity facility in Ho Chi Minh City, Vietnam, that provides care for both low- and high-risk pregnancies. Hung Vuong Hospital handles approximately 30,000–40,000 deliveries annually. The CS rate at the hospital has been increasing over time, nearly 47.6% [13]. Many expectant mothers choose Hung Vuong Hospital over district hospitals due to its highly specialized medical staff, advanced facilities, and perceived higher safety standards [14].

The sample size was determined using a formula for estimating a proportion. With an estimated proportion of CS preferences based on findings from a previous study of 18.8% [15], the marginal error of 3.5%, the participation rate of 90%, a minimum sample size required for this study was 533. Pregnant women beyond 36 weeks of gestation who attended routine antenatal care appointments were included in this study. Pregnant women under the age of 18 and those with anticipated medical or obstetrical complications from CS were excluded (Fig 1).

### Procedures and measurements

The trained research assistants recruited the participants and conducted face-to-face interviews to collect information through pre-defined questionnaires which included information about socio-demographic aspects, obstetrical factors, birth experiences, pregnant women's attitudes toward birth mode, and their awareness of the benefits and limitations of each birth method.

The outcome variable was women's preference for delivery mode (vaginal vs. cesarean birth) in the absence of medical indications. Independent variables included sociodemographic characteristics (e.g., age, residence, education, income, antenatal class attendance), obstetric history and current pregnancy status (e.g., parity, previous delivery mode, gestational age, estimated fetal weight), attitudes and expectations toward childbirth (e.g., fear of childbirth pain, belief in safety of CS, family influence, healthcare provider recommendation), and knowledge of benefits and drawbacks of different birth modes. Detailed definitions of all study variables are provided in the included questionnaire.

In our study, we defined healthy or low-risk gestation as pregnancy without any reported discomforts, high-risk medical conditions, or complications affecting the mother or fetus up to the time of the interview. We defined the variable mode of birth preference as the participant's self-reported choice of delivery method [16,17].

Knowledge about birth modes was evaluated based on 30 questions about the advantages and drawbacks of vaginal birth and CS. A scoring system was applied, whereby respondents who answered at least 25 out of 30 questions correctly were categorized as having "good knowledge." In contrast, those with fewer than 25 correct answers were considered to have "not good knowledge." We interviewed pregnant women while they were lying on a bed for a fetal heart rate monitor

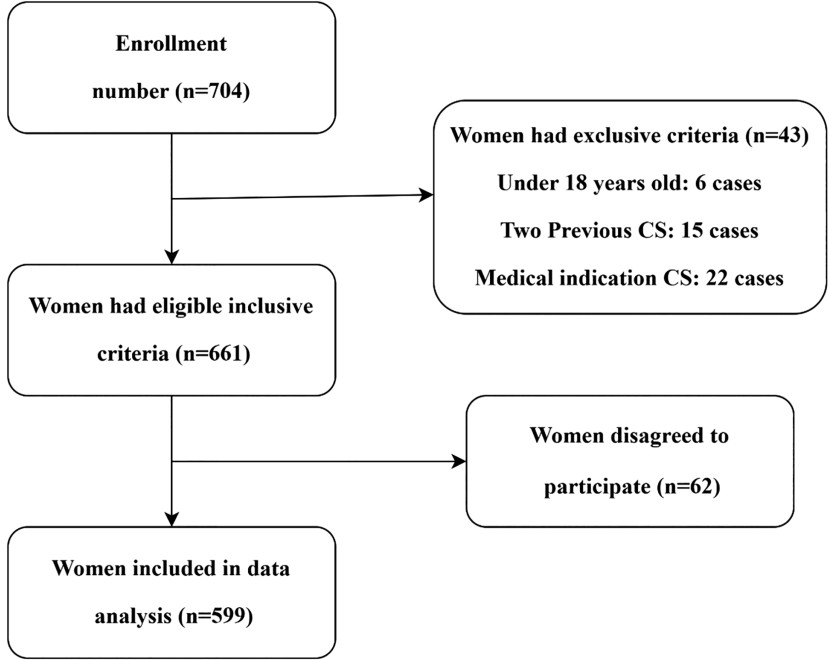

**Fig 1. Recruitment flowchart of pregnant women.**

in 30 minutes in the prenatal care unit. Other characteristics (e.g., estimated birth weight using ultrasound, gestational age at study participation) were obtained from the examination record on the interview day when prenant women did prenatal care. Technical aspects of the study including the use of the questionnaire were evaluated by the Council for Science and Technology of the study hospital, which was composed of obstetricians and gynecologists. A pilot study was then conducted among 30 pregnant women to test the questionnaire which resulted in a high level of internal reliability with a Cronbach's alpha of 0.804.

## Ethical approval

The Ethical Committee of Hung Vuong Hospital granted all the study procedures (approval No. 1684/HDDD-BVHV, dated April 10th, 2023). The study was conducted following the Declaration of Helsinki. Written informed consent was obtained from all participants before their participation in the study.

## Statistical analysis

The data were entered and stored using EpiData version 3.1, and statistical analyses were conducted using Stata version 17.0. Frequencies and percentages were used to summarize categorical variables. Binary logistic regression was utilized to calculate odds ratios (OR) and 95% confidence intervals (CI) for each variable. A backward elimination stepwise approach was used in multiple logistic regression with the initial inclusion of all variables with a p-value below 0.2 in the univariate analyses. The Hosmer Lemeshow test was used to evaluate the goodness of fit of the final model. Multicollinearity was assessed using the Variance Inflation Factor (VIF), and variables with VIF values above 5 were considered indicative of potential multicollinearity. Wald tests were also performed to assess the significance of individual predictors in the final model. A significance level of $p < 0.05$ was considered statistically significant.

## Results

Among the 661 pregnant women approached and recruited using a convenience sampling technique, 599 agreed to participate and completed the study, resulting in a participation rate of 90.6% (599/661).

The participants' ages ranged from 18 to 44, with a mean age of 29 (SD = 5.3) and 36.9% over 30 years old. Over half of the participants completed high school or higher education. The average gestational age at study participation was 37.6 weeks (SD = 1.2), and the average estimated birth weight using ultrasound was 2913 grams (SD = 392), ranging from 1700 to 4359 grams. Among the participants, 18.7% of prenatal fetuses had an estimated birth weight below 2500 grams, while the majority (81.3%) weighed 2500 grams or more. Regarding gestational age, 28.2% of the women were <37 weeks, 51.8% were between ≥37 and <39 weeks, and only 20.0% were at ≥39 weeks. A small percentage (18.4%) of participants attended antenatal classes at the hospital. Among the participants, 54.6% were nulliparous, while 45.4% were multiparous, with 62.5% of the latter having previously experienced vaginal birth. Our analysis revealed that 162 out of 599 participants (27.1%) preferred CS (Table 1).

In the bivariate analysis, there were no significant associations between the above characteristics and preference for CS, except for previous birth (Table 1). Compared to nulliparous women, those with a previous vaginal birth were less likely to prefer CS (OR = 0.47, 95% CI: 0.27–0.84; p = 0.001). In contrast, women with a history of CS birth were much more likely to prefer CS (OR = 18.67, 95% CI: 10.56–33.02; p < 0.001). When vaginal birth was used as the reference group, women with previous CS birth had significantly higher odds of preferring CS (OR = 39.32, 95% CI: 19.39–79.71). Among 162 (27.1%) women who preferred CS birth, their key concerns included complications related to vaginal birth (79.1%), the potential failure of vaginal birth (76.8%), and fears of childbirth pain (69.6%). Additionally, 41.2% of women expressed anxiety about postpartum sexual activities. Negative experiences regarding vaginal birth (37.4%) and advice from family members favoring CS (20.0%) were reported. A total of 44 women (7.3%) received recommendations from medical professionals for CS birth modes. Parity, gestational health, fears of natural childbirth, fear of labor pain, perceived safety for both mother and baby, family life considerations, and knowledge of the benefits and drawbacks of different birth modes were all significantly associated with preferences of CS (p < 0.05) (Table 2).

In multiple logistic regression analysis, multiparous women with previous CS were more likely to prefer CS than nulliparous women (adjusted Odds Ratio (aOR) = 41.48, 95% CI 17.56–97.99). Women with fear of complication from vaginal birth were over six times more likely to prefer CS (aOR = 6.27, 95% CI: 2.24–17.57). Participants who believed CS was safer for the infant demonstrated a strong preference for this mode (aOR = 5.08, 95% CI: 2.73–9.46). Those who believed that the date of birth impacts family life had a higher odds of preferences for CS (aOR = 3.48, 95% CI: 1.49–8.13). Additionally, advice from relatives significantly influenced preferences, as women who received such advice were 6.58 times more likely to opt for CS (aOR = 6.58, 95% CI: 3.24–13.37). Women who received recommendations from healthcare practitioners were significantly more likely to prefer CS, (adjusted OR = 7.15, 95% CI: 2.49–20.48). Moreover, women's preferences for mode of birth were significantly affected by exposing negative childbirth experiences of others with aOR = 2.66; 95% CI:1.43–4.98. Statistically significant association was found between labor companionship expectations and CS preference (aOR = 0.45; 95% CI: 0.24–0.85). Concern about postpartum sexual activities also appeared as a significant predictor, with women representing this concern being nearly twice as likely to prefer CS (aOR = 1.93; 95% CI: 1.04–3.56). Futhermore, knowledge of the benefits and drawbacks of different birth mode were likely to be critical determinant of preferences (aOR = 0.47; 95% CI: 0.26–0.86) (Fig 2). The VIF values for the final model ranged from 1.03 to 1.32, indicating no multicollinearity issues. Model fit was supported by a Pearson's Chi-squared test (p = 0.696) and a Hosmer-Lemeshow test (p = 0.748), both suggesting that the model fits the data well.

## Discussion

Vietnam has not established a specific law to regulate or restrict elective CS without medical indications. However, without proper control measures, the rising CS rates may cause increased maternal and neonatal health risks and place a more

**Table 1. Participant's sociodemographic characteristics and associated factors of preference for Cesarean section.**

| Characteristics | Total n (%) | Preference of mode of birth | | p | OR (95% CI) |
| --- | --- | --- | --- | --- | --- |
| | | Cesarean Section (n=162; 27.1%) n (%) | Vaginal Birth (n=437; 72.9%) n (%) | | |
| **Age** (year) | | | | | |
| < 25 | 120 (20.0) | 28 (17.3) | 92 (21.0) | | Ref |
| 25-30 | 258 (43.1) | 64 (39.5) | 194 (44.4) | 0.756 | 1.08 (0.65-1.80) |
| >30 | 221 (36.9) | 70 (43.2) | 151 (34.6) | 0.105 | 1.52 (0.92-2.53) |
| **Educational level** | | | | | |
| <High school | 369 (61.6) | 102 (63.0) | 267 (61.1) | | Ref |
| >High school | 230 (38.4) | 60 (37.0) | 170 (38.9) | 0.677 | 0.92 (0.64-1.34) |
| **Personal monthly income** | | | | | |
| <20 million VND | 358 (59.8) | 94 (58.0) | 264 (60.4) | | Ref |
| >20 million VND | 241 (40.2) | 68 (42.0) | 173 (39.6) | 0.597 | 1.10 (0.77-1.59) |
| **Residence** | | | | | |
| Urban | 384 (64.1) | 99 (61.1) | 285 (65.2) | | Ref |
| Suburban | 215 (35.9) | 63 (38.9) | 152 (34.8) | 0.352 | 1.19 (0.82-1.73) |
| **Body mass index** (kg/m$^2$) | | | | | |
| <25 | 204 (34.1) | 150 (34.3) | 54 (33.3) | | Ref |
| ≥25 | 395 (65.9) | 287 (65.7) | 108 (66.7) | 0.820 | 1.05 (0.71-1.53) |
| **Estimated birth weight using ultrasound (gram)** | | | | | |
| <2500 | 112 (18.7) | 33 (20.4) | 79 (18.1) | | Ref |
| ≥2500 | 487 (81.3) | 129 (79.6) | 358 (81.9) | 0.523 | 0.86(0.55-1.36) |
| **Gestational age at study participation (weeks)** | | | | | |
| <37 | 169 (28.2) | 53 (32.7) | 116 (26.5) | | Ref |
| ≥37-<39 | 310 (51.8) | 81 (50.0) | 229 (52.4) | 0.223 | 0.77 (0.51-1.17) |
| ≥39 | 120 (20.0) | 28 (17.3) | 92 (21.1) | 0.135 | 0.67 (0.39-1.14) |
| **Religion affiliation** | | | | | |
| No | 432 (72.1) | 117 (72.2) | 315 (72.1) | | Ref |
| Yes | 167 (27.9) | 45 (27.8) | 122 (27.9) | 0.973 | 0.99 (0.66-1.48) |
| **Antenatal class attendance** | | | | | |
| No | 489 (81.6) | 130 (80.2) | 359 (82.2) | | Ref |
| Yes | 110 (18.4) | 32 (19.8) | 78 (17.8) | 0.593 | 1.13 (0.72-1.79) |
| **Previous birth** | | | | | |
| Nulliparous | 327 (54.6) | 62 (38.3) | 265 (60.6) | | Ref |
| Vaginal birth | 170 (28.4) | 17 (10.5) | 153 (35.0) | **0.001** | 0.47 (0.27 - 0.84) |
| Cesarean section | 102 (17.0) | 83 (51.2) | 19 (4.3) | **<0.001** | 18.67 (10.56-33.02) |

*Bold values denote statistical significance at the p<0.05*

substantial burden on the healthcare system. Thus, understanding and managing factors that contribute to a high rate of CS is crucial for ensuring health and reducing complications for both mother and fetus. In our study, the women's CS preference rate was relatively high, and was associated with various factors. These associated factors included being multiparous with a history of CS birth, fear of complications from vaginal birth, safer for the baby, advice from relatives, healthcare provider recommendations, family support during labor, and concerns about postpartum sexual activities and knowledge of the benefits and drawbacks of mode of birth. Our findings emphasize the critical role of healthcare providers

**Table 2. Perception and knowledge about birth modes and preference for Cesarean section.**

| Characteristics | Total n (%) | Preference of mode of birth | | p | OR (95% CI) |
|---|---|---|---|---|---|
| | | Cesarean Section (n=162; 27.1%) n (%) | Vaginal Birth (n=437; 72.9%) n (%) | | |
| **Low risk gestation** | | | | | |
| No | 111 (18.5) | 52 (32.1) | 59 (13.5) | | Ref |
| Yes | 488 (81.5) | 110 (67.9) | 378 (86.5) | <0.001 | 0.33 (0.21-0.51) |
| **Fear of childbirth pain** | | | | | |
| No | 182 (30.4) | 33 (20.4) | 149 (34.1) | | Ref |
| Yes | 417 (69.6) | 129 (79.6) | 288 (65.9) | 0.001 | 2.02 (1.31-3.11) |
| **Fear of failure vaginal birth** | | | | | |
| No | 139 (23.2) | 14 (8.6) | 125 (28.6) | | Ref |
| Yes | 460 (76.8) | 148 (91.4) | 312 (71.4) | <0.001 | 4.24 (2.36-7.61) |
| **Fear of complication of vaginal birth** | | | | | |
| No | 125 (20.9) | 10 (6.2) | 115 (26.3) | | Ref |
| Yes | 474 (79.1) | 152 (93.8) | 322 (73.7) | <0.001 | 5.43 (2.77-10.66) |
| **Cesarean section has a less pain than vaginal birth** | | | | | |
| No | 411 (68.6) | 74 (45.7) | 337 (77.1) | | Ref |
| Yes | 188 (31.4) | 88 (54.3) | 100 (22.9) | <0.001 | 4.01 (2.74-5.87) |
| **Safer for the mother** | | | | | |
| No | 449 (75.0) | 76 (46.9) | 373 (85.4) | | Ref |
| Yes | 150 (25.0) | 86 (53.1) | 64 (14.6) | **<0.001** | 6.59 (4.39-9.91) |
| **Safer for the baby** | | | | | |
| No | 436 (72.9) | 64 (39.8) | 372 (85.1) | | Ref |
| Yes | 162 (27.1) | 97 (60.2) | 65 (14.9) | **<0.001** | 8.67 (5.75-13.09) |
| **Date of birth affect to the family's life** | | | | | |
| No | 528 (88.1) | 118 (72.8) | 410 (93.8) | | Ref |
| Yes | 71 (11.9) | 44 (27.2) | 27 (6.2) | **<0.001** | 5.66 (3.36-9.53) |
| **Choose date of birth** | | | | | |
| No | 551 (92.0) | 127 (78.4) | 424 (97.0) | | Ref |
| Yes | 48 (8.0) | 35 (21.6) | 13 (3.0) | **<0.001** | 8.99 (4.61-17.51) |
| **Relative's advice for Cesarean section** | | | | | |
| No | 479 (80.0) | 77 (47.5) | 402 (92.0) | | Ref |
| Yes | 120 (20.0) | 85 (52.5) | 35 (8.0) | **<0.001** | 12.68 (7.98-20.15) |
| **Recommendation of healthcare provider for Cesarean section** | | | | | |
| No | 555 (92.7) | 128 (79.0) | 427 (97.7) | | Ref |
| Yes | 44 (7.3) | 34 (21.0) | 10 (2.3) | **<0.001** | 11.34 (5.45-23.59) |
| **Allows better control of time of birth** | | | | | |
| No | 405 (67.6) | 86 (53.1) | 319 (73.0) | | Ref |
| Yes | 194 (32.4) | 76 (46.9) | 118 (27.0) | **<0.001** | 2.39 (1.64-3.47) |
| **Exposing negative experience of other women** | | | | | |
| No | 375 (62.6) | 66 (40.7) | 309 (70.7) | | Ref |
| Yes | 224 (37.4) | 96 (59.3) | 128 (29.3) | **<0.001** | 3.51 (2.41-5.11) |
| **Labor companionship expectation** | | | | | |
| No | 199 (33.2) | 62 (38.3) | 137 (31.4) | | Ref |
| Yes | 400 (66.8) | 100 (61.7) | 300 (68.6) | 0.111 | 0.74 (0.51-1.07) |

*(Continued)*

**Table 2.** (Continued)

| Characteristics | Total n (%) | Preference of mode of birth | | p | OR (95% CI) |
|---|---|---|---|---|---|
| | | Cesarean Section (n=162; 27.1%) n (%) | Vaginal Birth (n=437; 72.9%) n (%) | | |
| **Concerns about postpartum sexual activities** | | | | | |
| No | 352 (58.8) | 61 (37.7) | 291 (66.6) | | Ref |
| Yes | 247 (41.2) | 101 (62.3) | 146 (33.4) | **<0.001** | 3.30 (2.27-4.80) |
| **Knowledge of the benefits and drawbacks of mode of birth** | | | | | |
| Not good | 271 (45.2) | 93 (57.4) | 178 (40.7) | | Ref |
| Good | 328 (54.8) | 69 (42.6) | 259 (59.3) | **<0.001** | 0.51 (0.35-0.73) |

*Bold values denote statistical significance at the p<0.05*

in creating women's preferences for cesarean section. While recommendations from clinicians, family advice, and fear of vaginal birth complications strongly increased CS preference, other factors such as good knowledge about delivery modes and the expectation of labor companionship acted as protective influences, reducing women's likelihood of preferring CS. Interventions should therefore not only target clinicians through education, guideline reinforcement, and shared decision-making training, but also strengthen the protective factors through enhanced prenatal education, counseling, and supportive care during labor to decrease unnecessary CS births [1].

The proportion of women who preferred CS vary significantly across different countries worldwide. For example, the relatively high rate of CS preference in our study is consistent with previous studies, such as findings from studies in Ethiopia (28.9%) and China (23.3%) [18,19]. Moreover, our rate is slightly higher than one study carried out in a general hospital in Vietnam in 2013 (18.1%). However, our rate is much higher than in developed countries such as European countries, where the preference for CS ranges from 3.5% in primiparous women to 8.7% in multiparous women, or 7.6% in Iceland and 18.4% in Australia [20,21]. It is noteworthy that EU countries with strong midwifery-led models, such as the Netherlands and Sweden, report lower CS rates and perinatal mortality, possibly due to more continuity of care [22,23,24]. However, this rate is still lower than those reported in studies conducted in Slovakia and Egypt [17]. This discrepancy provides valuable context for our findings within the global landscape of CS rates. The differences in the rates reported reflect differences in health systems, available resources, cultural and social attitudes, and health and legal policies in various regions. This diversity underscores the significance of conducting localized research to comprehend contextual and cultural elements influencing CS rates. Understanding these factors can help healthcare systems tailor their services to meet the specific needs of their population, potentially reducing unnecessary CS rates and improving overall maternal and fetal health. In Vietnam, the National Strategy for the Protection, Care, and Improvement of People's Health for the Period up to 2030, with a Vision to 2045, aims to develop a modern and internationally integrated healthcare system, enhance the quality of medical services, and achieve universal health coverage, with a particular focus on maternal and reproductive health. Additionally, the National Guidelines on Reproductive Health Services provide comprehensive guidance on reproductive healthcare, covering maternal care before, during, and after childbirth, family planning, and the prevention of sexually transmitted infections (STIs) [25].

Our findings on the correlates of women's preference for CS are consistent with previous studies. For example, a systematic review of Agustina Mazzoni on 38 studies revealed that women who had previously undergone a CS indicated a higher preference for CS than women who had not (29.4% versus 10.1%) [26]. Futher, fear of complication from vaginal birth was one of feelings of fear of childbirth about uncertainty and anxiety before, during or after childbirth [27]. A Hege Therese's study in Norway with 1789 women demontrated that preference for elective CS was significantly corelated with fear of childbirth (aOR = 4.6, 95% CI: 2.9–7.3) [28]. Likewise, a study of Huijuan Liang also pointed to most people

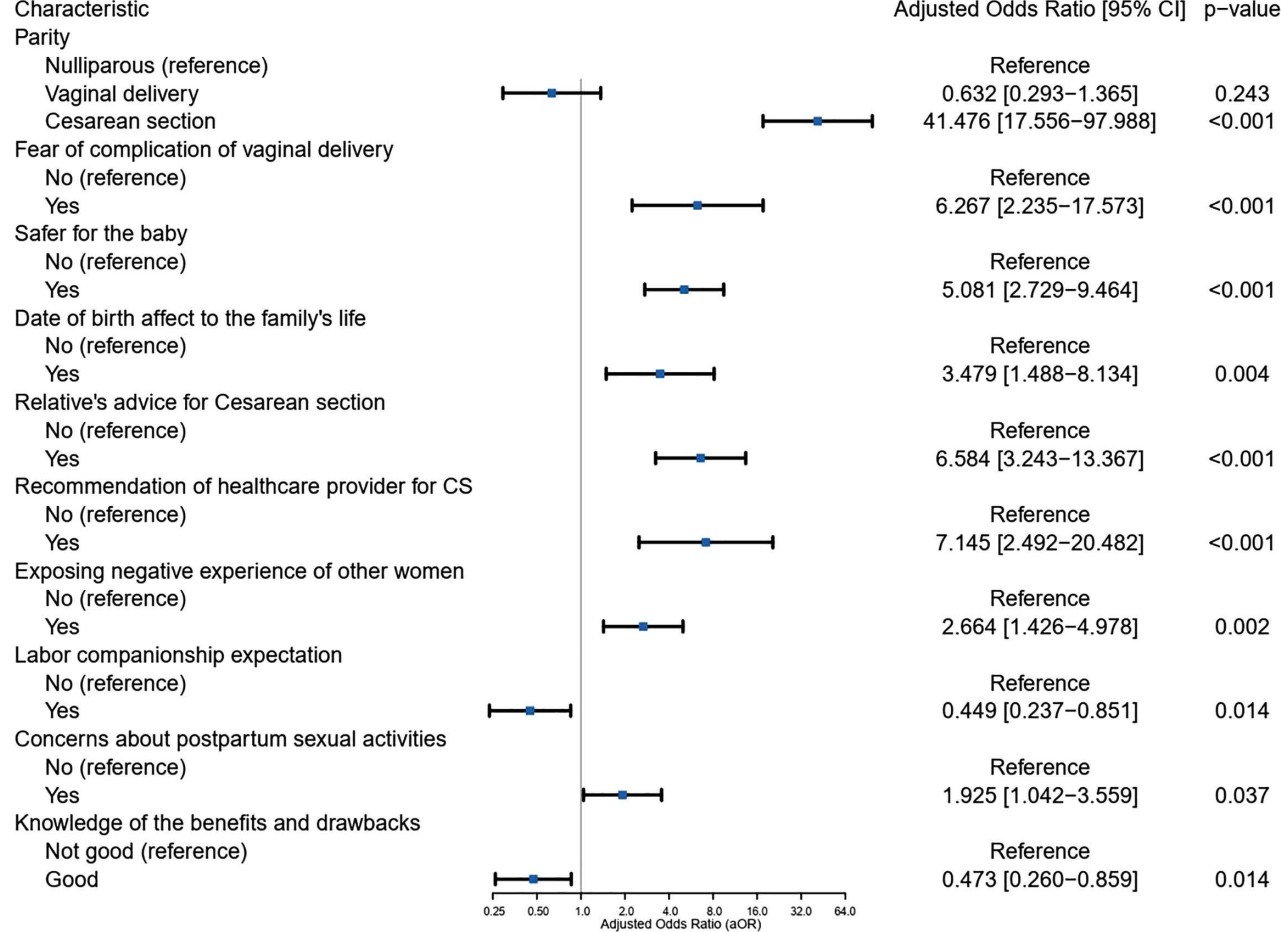

| Characteristic | Adjusted Odds Ratio [95% CI] | p-value |
|---|---|---|
| Parity | | |
| Nulliparous (reference) | Reference | |
| Vaginal delivery | 0.632 [0.293–1.365] | 0.243 |
| Cesarean section | 41.476 [17.556–97.988] | <0.001 |
| Fear of complication of vaginal delivery | | |
| No (reference) | Reference | |
| Yes | 6.267 [2.235–17.573] | <0.001 |
| Safer for the baby | | |
| No (reference) | Reference | |
| Yes | 5.081 [2.729–9.464] | <0.001 |
| Date of birth affect to the family's life | | |
| No (reference) | Reference | |
| Yes | 3.479 [1.488–8.134] | 0.004 |
| Relative's advice for Cesarean section | | |
| No (reference) | Reference | |
| Yes | 6.584 [3.243–13.367] | <0.001 |
| Recommendation of healthcare provider for CS | | |
| No (reference) | Reference | |
| Yes | 7.145 [2.492–20.482] | <0.001 |
| Exposing negative experience of other women | | |
| No (reference) | Reference | |
| Yes | 2.664 [1.426–4.978] | 0.002 |
| Labor companionship expectation | | |
| No (reference) | Reference | |
| Yes | 0.449 [0.237–0.851] | 0.014 |
| Concerns about postpartum sexual activities | | |
| No (reference) | Reference | |
| Yes | 1.925 [1.042–3.559] | 0.037 |
| Knowledge of the benefits and drawbacks | | |
| Not good (reference) | Reference | |
| Good | 0.473 [0.260–0.859] | 0.014 |

**Fig 2. Factors independently associated with women's CS preference in the multiple logistic regression.**

preferred CS because they thought it was better for infants (24.3%) [29]. In addition, a survey conducted in four tertiary hospitals in China involving 1,169 pregnant women found that 68% of women whose husbands preferred CS also chose CS as their birth mode [29]. Similarly, a survey in Egypt reported that 20.4% of women who received healthcare provider recommendations for CS opted for CS [30]. Moreover, according to Ingela Wiklund's survey of 357 pregnant women in Sweden reported a considerable influence of lack of family support during labor on the desire for a CS (p < 0.001) [31]. Notably, our study's results align with Demsar's research in Slovenia, which involved 191 pregnant women and found that 7.8% of them preferred CS due to concerns about their future sexual lives [32]. Furthermore, Rami H Al-Rifai's survey in the United Arab Emirates, which included 1,617 pregnant women, indicated that those who opted for CS were more likely to have insufficient knowledge about birth modes [33]. Interestingly, our study reveals a significant correlation between women's preferences for CS and their exposure to negative childbirth experiences of others. In Vietnam, folk beliefs suggest that a child's destiny and future are influenced by their birth time and date. This practice has significantly contributed to the rising cesarean section rates, particularly in urban areas like Ho Chi Minh City and Hanoi [12,34].

However, our study did not find statistically significant associations between Vietnamese women's choice of birth mode and other factors such as age. Previous studies have also reported inconsistent findings about the associated factors. For example, a survey by Ruibin Deng in China involving 1,283 pregnant women found that women aged 30 were 4.29 times

more likely to prefer CS than those under 25 [19]. Similarly, Annika Karlstrom's study in Sweden, which involved 1,212 women, revealed primiparous women aged 35 or older were less likely to choose CS [35]. In contrast, Zahide Kosan's research on 418 individuals in Turkey found no statistically significant correlation between birth preference and age [36]. The disparity in findings between our study and previous research indicates that specific variables or subgroups are more likely than others to select CS, but this preference depends on the context of each locality. Therefore, studies exploring the specific factors of each context are necessary to accurately identify the group at risk of excessive use of CS.

Our study has several limitations. First, this study was conducted at a single hospital, which may limit the generalizability of the findings to other regions or healthcare settings in Vietnam. Second, the study design captures participants' preferences and influencing factors at only one point of time, which may not reflect changes over time. Third, the sample size constraints reduced the statistical power of the study, restricting the ability to conduct subgroup analyses (e.g., by cesarean section history or parity) and resulting in wide confidence intervals in regression models, which may affect the precision and robustness of the findings. Fourth, although estimated fetal weight was collected, we did not assess estimated fetal weight percentiles, which could better reflect fetal growth and warrant further study. Finally, the study relied on self-reported questionnaires, which could introduce recall or social desirability bias. Although we cross-validated the interview data with prenatal examination records to mitigate this, further research is essential to gain a more comprehensive understanding of the critical factors influencing Vietnamese women's preferences for the mode of birth.

## Conclusions

This study reveals a high rate of preference for CS among Vietnamese women, shaped by multiple factors including prior cesarean birth, fear of vaginal birth complications, perceived safety for the baby, advice from relatives, recommendations from healthcare providers, availability of family support during labor, concerns about postpartum sexual health, and women's awareness of the advantages and disadvantages of different birth modes. The findings underscore the importance of improved prenatal education and counseling to manage medical and non-medical factors, ensuring that women can make informed decisions about their birth options, a necessity for their overall health and well-being. Especially, interventions targeting clinician counselling and practice are crucial to reduce unnecessary cesarean deliveries.

## Supporting information

**S1 Data. Data Questionnaire Raw Full.**
(ZIP)

## Acknowledgments

We thank all the data collectors for their valuable contributions to this study. The authors are also grateful for the support from Hung Vuong Hospital and to all the participants who contributed to it.

## Author contributions

**Conceptualization:** Trang Thi Thuy Hoang.

**Data curation:** Trang Thi Thuy Hoang, Hue Thi Hoang, Dan Trieu Thanh Nguyen, Hang Thi Phan.

**Methodology:** Trang Thi Thuy Hoang, Chau Giang Huynh, Truc Thanh Thai.

**Project administration:** Trang Thi Thuy Hoang, Hang Thi Phan.

**Resources:** Hue Thi Hoang, Dan Trieu Thanh Nguyen, Hang Thi Phan.

**Software:** Trang Thi Thuy Hoang.

**Supervision:** Trang Thi Thuy Hoang, Hang Thi Phan.

**Validation:** Oanh Thi Hoang Trinh, Chau Giang Huynh, Truc Thanh Thai, Quan Minh Pham, Minh Thien Nguyen.

**Visualization:** Trang Thi Thuy Hoang, Chau Giang Huynh, Truc Thanh Thai, Quan Minh Pham, Minh Thien Nguyen.

**Writing – original draft:** Trang Thi Thuy Hoang.

**Writing – review & editing:** Trang Thi Thuy Hoang, Oanh Thi Hoang Trinh, Chau Giang Huynh, Truc Thanh Thai, Quan Minh Pham, Hue Thi Hoang, Dan Trieu Thanh Nguyen, Minh Thien Nguyen, Hang Thi Phan.

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
