## [Decision Letter · Decision Letter 0]

18 Feb 2025

Thank you for submitting your manuscript to PLOS ONE. After careful consideration, we feel that it has merit but does not fully meet PLOS ONE’s publication criteria as it currently stands. Therefore, we invite you to submit a revised version of the manuscript that addresses the points raised during the review process.

We look forward to receiving your revised manuscript.

Kind regards,

Kornelia Zaręba, MD

Academic Editor

PLOS ONE

**Journal Requirements:**

1. When submitting your revision, we need you to address these additional requirements.Please ensure that your manuscript meets PLOS ONE's style requirements, including those for file naming. The PLOS ONE style templates can be found at https://journals.plos.org/plosone/s/file?id=wjVg/PLOSOne_formatting_sample_main_body.pdf and https://journals.plos.org/plosone/s/file?id=ba62/PLOSOne_formatting_sample_title_authors_affiliations.pdf 2. Thank you for stating the following financial disclosure:  Hung Vuong Hospital funded this study (Decision No. 7410/QĐ-BVHV, dated December 15th, 2023)Please state what role the funders took in the study.  If the funders had no role, please state: "The funders had no role in study design, data collection and analysis, decision to publish, or preparation of the manuscript." If this statement is not correct you must amend it as needed. Please include this amended Role of Funder statement in your cover letter; we will change the online submission form on your behalf. 3. In the online submission form, you indicated that Data generated or analyzed during this study are available from the corresponding author upon reasonable request. Data available under certain conditions.All PLOS journals now require all data underlying the findings described in their manuscript to be freely available to other researchers, either a. In a public repository, b. Within the manuscript itself, or c. Uploaded as supplementary information.This policy applies to all data except where public deposition would breach compliance with the protocol approved by your research ethics board. If your data cannot be made publicly available for ethical or legal reasons (e.g., public availability would compromise patient privacy), please explain your reasons on resubmission and your exemption request will be escalated for approval.

Reviewers' comments:

Reviewer's Responses to Questions

**Comments to the Author**

1. Is the manuscript technically sound, and do the data support the conclusions?

Reviewer #1: Yes

Reviewer #2: Partly

2. Has the statistical analysis been performed appropriately and rigorously?

Reviewer #1: No

Reviewer #2: No

3. Have the authors made all data underlying the findings in their manuscript fully available?

Reviewer #1: No

Reviewer #2: Yes

4. Is the manuscript presented in an intelligible fashion and written in standard English?

Reviewer #1: No

Reviewer #2: Yes

**Reviewer #1:**  Dear authors,

I have read your article with high interest and I could find some points that deserves attention.

Title, for example, i suggest you to modify it since your sample doesn't represent Vietnamese Women. I could not find the regional origin of the participants of this study, but i'm pretty sure they don't represent all women population.

In the introduction this should also be clarified that you consider only women followed by a specific hospital.

Methods:

For example, in settings you should mention the reason for choosing this hospital, and it would be great if you could describe a little bit about the prenatal coverage in the country. Can the antenatal care be made in primary healthcare networks? How accessible is the antenatal care for the population? Do people living in rural zone have easily access to antenatal care? I was also wondering in what gestational age most women started the prenatal in your country. Is there any regional discrepancies in CS and in prenatal rates?

Another point was about the multiple regression. Although you briefly describe it, I was not sure if you used stepwise,..., and what was the VIF considered. Additionally, I advice you to do the Wald-test to bring more robust results.

From where you did take the "estimated proportion of CS preferences of 18.8%"?

About questionary, was it evaluated by specialists before pilot test? I really was questioning myself about the origin of these voluntaries.

I see you also use descriptive analysis (n and %), but none was told in methods.

In discussion it would be great if you could discuss about public policies in your country regarding the CS combat. Is there any specific law? Is there any public policy towards women's health? What about cultural aspects? Because of that the origin of the participants is really important.

In the study limitation should be included that your sample doesn't represent the national women population.

Minor point: The wording of the text needs to be revised.

**Reviewer #2:**  Thank you for the opportunity to review this paper which is related to the important issue of increasing CS rates in many countries.

This was a cross-sectional study of 599 pregnant women who received care at a single hospital in Ho Chi Minh City, Vietnam in 2023 to assess their preferences for birth mode.

They found 27% of women would prefer a CS with the main reasons including health practitioner recommendations and as well as different non-medical reasons.

Disclosure statements

Ethics statement is incomplete. Please indicate the type of consent obtained from the participant (as per Methods section).

General: I suggest using ‘birth mode’ instead of ‘delivery mode’ throughout the paper as this is more empowering for women.

Abstract: please define all abbreviations on first use (e.g. Odds ratio, confidence interval)

Introduction

Please categorize the potential risks of CS into maternal and fetal/infant risks. I also suggest adding the implications for later pregnancies (recurrent CS, uterine rupture).

Methods: Please add more details about the recruitment. Were all women who met the inclusion criteria or just a sample (random, convenience)? What was the participation fraction? These could also be added to Figure 1.

I suggest including the data collection instrument as supplementary material. Please describe how the birth characteristics (e.g. birth weight, gestation) were obtained. Were information about parity and birth mode for previous births obtained from the mother or her medical records? While this is mentioned in limitations, it should be described in methods first.

Please define ‘healthy gestation’ (item in Figure 2).

Statistical Methods.

Please describe in more detail which variables were tested in the multivariable logistic models and list those included in the final models ( and list in a footnote in Figure 2).

I suggest doing sensitivity analyses by parity to compare the views of nulliparous women to multiparous women, and, among multiparous women compare women who had a previous CS to those who did not. This would allow the authors to better compare their findings to previous literature.

Results

Rather than giving mean maternal age, infant birth weight and gestation, I suggest showing these as categorical variables in Table 1. Was the reason for previous CS obtained from multiparous women who had had one?

Table 1:

As the mean maternal age was 27 years and only a small percentage of women were ages over 40, it would be better to use different categorisation for maternal age (e.g <25, 25-29, 30-34 and 35 or more years). If the authors wish to use only 2 categories, a good choice is to split at the median value. The current categorisation may be the reason why no association was found between birth mode preference and maternal age.

Please also use more standard cut points for BMI. In my opinion, P values are not necessary in the table as the confidence intervals are sufficient. Please indicate the Reference group instead of OR = 1 ( as used in Figure 2).

Discussion

As the authors mentioned, differences in health care systems may explain differences in both CS rates and women’s preferences. to giv ethe reader more context, I suggest describing the health care system in Vietnam and particularly more details about Hung Vuong Hospital in the Introduction/Methods so the findings can be compared to similar settings. Please give details about the types of health care practitioners, whether Hung Vuong Hospital is a public or private hospital, level of care provided and how it compares to other birthing sites in Vietnam. Please also discuss the views of health providers regarding CS in Vietnam and describe the relevant guidelines for recommending a CS on medical grounds.

**Do you want your identity to be public for this peer review?** For information about this choice, including consent withdrawal, please see our Privacy Policy

Reviewer #1: No

Reviewer #2: No

---

## [Author Response · Author response to Decision Letter 1]

4 Apr 2025

Please see our responses in the file attached. Many thanks.

---

## [Decision Letter · Decision Letter 1]

9 Jun 2025

Dear Dr. Thai,

Thank you for submitting your manuscript to PLOS ONE. After careful consideration, we feel that it has merit but does not fully meet PLOS ONE’s publication criteria as it currently stands. Therefore, we invite you to submit a revised version of the manuscript that addresses the points raised during the review process.

**Please respond to all reviewers comments**

We look forward to receiving your revised manuscript.

Kind regards,

Ahmed Mohamed Maged, MD

Academic Editor

PLOS ONE

Reviewers' comments:

Reviewer's Responses to Questions

**Comments to the Author**

Reviewer #2: (No Response)

Reviewer #3: (No Response)

Reviewer #4: (No Response)

Reviewer #5: (No Response)

2. Is the manuscript technically sound, and do the data support the conclusions?

Reviewer #2: Partly

Reviewer #3: Partly

Reviewer #4: Partly

Reviewer #5: Partly

3. Has the statistical analysis been performed appropriately and rigorously?

Reviewer #2: Yes

Reviewer #3: I Don't Know

Reviewer #4: Yes

Reviewer #5: Yes

4. Have the authors made all data underlying the findings in their manuscript fully available?

Reviewer #2: Yes

Reviewer #3: Yes

Reviewer #4: Yes

Reviewer #5: Yes

5. Is the manuscript presented in an intelligible fashion and written in standard English?

Reviewer #2: No

Reviewer #3: Yes

Reviewer #4: Yes

Reviewer #5: Yes

**Reviewer #2:**  Thank you for addressing most of my queries.

Re the use of birth mode vs delivery mode. There are still many references to delivery (e.g Table 1 header vaginal delivery). I suggest doing a search of the document for ‘delivery’ or ‘delivering’ and replacing all with birth or birthing.

Introduction

Page 4 Line 55;

Please clarify that you are referring to ‘respiratory issues in the infant’.

Page 5, line 118: The authors claim that birth weight and gestation were collected were collected at an antenatal visit. Please correct this since these would only be available after the birth.

Table 1:

I suggest replacing ‘healthy’ pregnancy with the more commonly used term ‘low risk’.

Please classify gestation using the standard groupings (preterm <37 weeks, early (37-38 weeks and late term (39+ weeks).

Discussion.

The authors highlight the low CS rates in many EU countries. I suggesting adding a comment about the importance of midwifery models of care in these countries and corresponding low rates of perinatal death.

Please discuss the small proportion of births of >38 weeks gestation. There is a large literature about the long-term consequences for the infant of preterm and early term birth (37-38 weeks gestation). The Discussion focuses on the need for improved antenatal education. Please discuss if there is also a need for education of clinicians.

In the response, the authors stated that the study was insufficiently powered to do subgroup analyses stratified by parity and history of CS. Please add this to the study limitations.

**Reviewer #3: ** Thank you for the opportunity to review this revised manuscript on a cross-sectional study of factors influencing birth mode preferences among pregnant women at a hospital in Vietnam. The authors have strengthened the manuscript in response to previous review comments, but further clarifications are needed.

My main concern with technical soundness of this manuscript is that it is not clear how the study variables were defined or assessed. It is good that the questionnaire is attached as a supplementary file, but each of the variables still need to be clearly defined in the manuscript, with details of assessment and scoring methods. For example, how was ‘healthy gestation’ defined for participants? Does the result in Table 2 mean that the participant had a healthy pregnancy (yes/no) or that they believe that a certain birth mode (vaginal/c-section) is preferable after a healthy pregnancy? As another example, how were data on ‘knowledge of benefits and drawbacks’ categorized into a score of ‘good’ or ‘not good’ (Fig 2)? For the outcome variable, how was ‘preference’ defined, and how does it relate to whether a c-section was actually booked for the participant?

Please also clarify in the manuscript that the birth weight and gestational age data refer to the previous pregnancy. I was confused about this until reading the response to previous reviewers’ comments.

The wide confidence intervals for many of the regression analysis results are a concern, which should be addressed in the Discussion. Some corrections are also needed: Line 154 is not accurate (there were no significant associations between the above noted characteristics, based on Table 1), and in Figure 2, the scale needs to have even increments to accurately portray the results including the spread of CIs. Comparisons of c-section rates in the Introduction need to distinguish between national data and site-specific studies. Instead of providing more comparisons in the Discussion, it would be better to compare this study’s finding with the hospital and national rates, to discuss the generalizability of the sample. It should also be noted that this study collected preference data, not c-section rate.

Regarding statistical analysis, the methods appear appropriate but it is not clear which variables were retained in the final model after applying backwards elimination. Fig 2 includes aOR results for ‘labour companion’s expectations’ but this variable had p>0.05 in Table 2. It should not have been included in the adjusted model, according to the method described in lines 135-136.

The manuscript is generally well written. Please clarify line 57: I am not sure what ‘using CS in clinical settings’ means. For lines 97-100, please include a reference for the first sentence, and delete the second sentence. Please provide a reference for the statement about cultural beliefs contributing to rising c-section rates (lines 255-256).

**Reviewer #4:**  in the method, it said cross sectional study. it means that the data will be from at least 2 different hospital. but, it only show from 1 hospital. so, please revise that the study just conduct for 1 specific hospital. Also, for Perception and knowledge about birth modes and preference for Cesarean section it show that 92% answer Recommendation of healthcare provider for Cesarean section. It need deeper discussion, how in Vietnam healthcare provider more recommend for cesarean section.

**Reviewer #5: ** Preferences for Cesarean Section Among Pregnant Women at a Terciary Hospital in Ho Chi Minh City, Vietnam: Influencing Factors and Implications for Prenatal Care

Cesarean rates worldwide, including in Vietnam, are high and growing, and this paper explores birthing preferences, in a sample of Vietnamese women. The topic is relevant, as many hospitals seek to de-medicalize childbirth by offering options to ease the stress of the mother and therefore improve mother and child outcomes. The sample is a good size to achieve this aim in a cross-sectional survey, and the data analysis methods are consistent with standard procedure.

Abstract

Include the current worldwide rate and the rate of CS in Vietnam in the first few sentences.

Methods

Setting and participants

On line 96, it sounds like the hospital may be private, could you confirm? What is the SES of the patients who attend?

Line 108- do you know why some declined participating in the study?

Line 112- Who recluted the participants, and who delivered the questionnaires?

Line 115- should say “were evaluated” rather than “was evaluated”

Line 118- you mention the medical record extraction was done on the day of the interview which was during a prenatal visit, but in results you present clinical data from the birth, such as week of gestation at birth, and baby weight. This is very confusing, especially since on Line 147 you talk about the participants as if they are infants. Did you do a later record extraction to gather these data? Or are these previous born children of the multiparous women?

Results

Also, it looks from Table 1 that no characteristics were associated with CS preference aside from previous CS experiences. If this is correct, you want to add a sentence to state that.

Somewhere in the results section, maybe where you list birth data (child characteristics), report how many wanted versus how many ended up having a CS.

Lines 155-160- the ordering of these results are confusing and I had to look back and forth to the table several times to understand that the first two sentences in this paragraph reflect the data in the table (nulliparous women set as reference group). I think this needs to be flushed out in the discussion section, and maybe simplified in the results. It is not necessary to repeat exactly the data in the table, so maybe summarize more succinctly the table data (without the numbers) and keep the sentence comparing past vaginal delivery group to past CS group.

Line 165- should be “among … women WHO preferred CS delivery”. Add the word “who”

Line 178- this part mentions multiple logistic regression, but what variables were in the model? In the Statistical analysis section of the Methods it says you use a back elimination stepwise approach with initial inclusion of all variables with p<.05, but in your Table 1 it looks like that would only be “previous delivery”. Please list variables included in adjusted model.

Below I copied and pasted the two similar sentences from the univariate and multivariate analysis. It looks like after adjusting, women with previous CS changed from being less likely to being much more likely to prefer CS. If I follow your methods, you adjusted for “previous delivery”, so using it as a predictor variable doesn’t make sense. Please clarify.

“Compared to nulliparous women, those with a previous CS

delivery were less likely to prefer CS (OR = 0.47, 156 95% CI: 0.27–0.84; p = 0.001).”

“Multiparous women with previous CS were more likely to prefer CS than nulliparous women (adjusted Odds Ratio (aOR) = 41.48, 95% CI 180 17.56 - 97.99).”

Discussion

If I am understanding the results right, only 7.4% of women discussed birth mode with their provider. That is surprising to me and would be good to take up in the discussion. Also, note in the table this figure is 7.3%. Please make sure all numbers in the table are reflected correctly in the text.

There is a document with track changes included at the end of the manuscript. I assume this was an error, but please clarify.

Line 252- you already mentioned this belief, so simply refer back to it rather than repeating it

**Do you want your identity to be public for this peer review?** For information about this choice, including consent withdrawal, please see our Privacy Policy

Reviewer #2: No

Reviewer #3: No

Reviewer #4: No

Reviewer #5: No

---

## [Author Response · Author response to Decision Letter 2]

31 Jul 2025

Please see our responses in the file attached.

---

## [Decision Letter · Decision Letter 2]

24 Aug 2025

Dear Dr. Thai,

Thank you for submitting your manuscript to PLOS ONE. After careful consideration, we feel that it has merit but does not fully meet PLOS ONE’s publication criteria as it currently stands. Therefore, we invite you to submit a revised version of the manuscript that addresses the points raised during the review process.

**ACADEMIC EDITOR: Please respond to all reviewers comments**

We look forward to receiving your revised manuscript.

Kind regards,

Ahmed Mohamed Maged, MD

Academic Editor

PLOS ONE

Journal Requirements:

Reviewers' comments:

Reviewer's Responses to Questions

**Comments to the Author**

Reviewer #2: (No Response)

Reviewer #3: (No Response)

Reviewer #4: (No Response)

Reviewer #5: All comments have been addressed

2. Is the manuscript technically sound, and do the data support the conclusions?

Reviewer #2: Partly

Reviewer #3: Partly

Reviewer #4: Partly

Reviewer #5: Yes

3. Has the statistical analysis been performed appropriately and rigorously?

Reviewer #2: Yes

Reviewer #3: Yes

Reviewer #4: Yes

Reviewer #5: Yes

4. Have the authors made all data underlying the findings in their manuscript fully available?

Reviewer #2: Yes

Reviewer #3: Yes

Reviewer #4: Yes

Reviewer #5: Yes

5. Is the manuscript presented in an intelligible fashion and written in standard English?

Reviewer #2: (No Response)

Reviewer #3: Yes

Reviewer #4: Yes

Reviewer #5: Yes

Reviewer #2: Revision 2

Thank you for addressing the majority of my comments.

The authors have clarified that information about fetal size and gestation were collected at the time of study participation. However, the categorisation does not discriminate well since 80% of participants fit into the same group for both measures .

I strongly suggest to separate those <37 weeks (i.e preterm at the time) from those at 37-38 weeks. It would also be more informative to use estimated fetal weight percentiles to indicate appropriateness of growth.

Discussion

Since recommendations from clinicians were such a powerful predictor of CS preference, the importance of measures targeting clinicians (page 11, line 232, marked version), should be reiterated in the Abstract and Conclusions as well.

Reviewer #3: The authors have made important clarifications and corrections to the manuscript in response to previous review comments. Thank you for this.

There is still an outstanding issue with the definition of variables. A previous review comment requested the authors to clearly define all included variables, but this was only done for certain variables. It is still unclear what others mean, and the attached study questionnaire does not provide any further details on how the questions were asked or the meaning of the items, especially for Section C. Please clearly define all variables in the Procedures and Measurements section. It is clearest for the reader if this is organized with the outcome variable first, then define the other study variables in the order in which they are presented in the data tables.

I see also that the Results section does not include data for several items that are in the questionnaire. Please clarify why these were left out, or provide the complete frequency data in Tables 1 and 2.

Please also ensure that the direction of association is acknowledged when presenting the regression results. Line 36 in the Abstract refers to “predictors” but some of the factors were protective against c-section preference. Lines 220-223 in the Discussion also need to clarify this.

Line 75 – Please clarify “has been increasing by approximately 47.6%” – is this per year or over some other timeframe? It would be clearer to provide the actual rates instead of % increase, if those data are available.

Lines 113-116, including Figure 1, should be placed at the beginning of the Results section.

Lines 223-227 – This statement about healthcare providers was added in response to another reviewer’s comments. It is an important point but family influence was also significant, and no other interventions are mentioned until the Conclusion. It would be better to highlight recommendations for both family/maternal education (lines 305-307) and healthcare provider education in the same place.

Line 234 – Please reword “It notes” to “It is noteworthy”

Line 257 – I believe the sentence should end after the word ‘childbirth’, then start a new sentence about the Norwegian study, with its citation.

Reviewer #4: please use the right term if the study only in one hospital (that's not cross sectional studies). remove the word cross sectional studies. it can define as case study in vietnam hospital because the data only in few month in 2023

Reviewer #5: Thank you for your thorough responses. I have reviewed your revised manuscript and you have addressed all of my concerns.

**Do you want your identity to be public for this peer review?** For information about this choice, including consent withdrawal, please see our Privacy Policy

Reviewer #2: No

Reviewer #3: No

Reviewer #4: No

Reviewer #5: No

---

## [Author Response · Author response to Decision Letter 3]

3 Oct 2025

Please see our responses in the file attached. Many thanks.

---

## [Editor Report · Decision Letter 3]

6 Oct 2025

Preferences for Cesarean Section Among Pregnant Women at a Tertiary Hospital in Ho Chi Minh City, Vietnam: Influencing Factors and Implications for Prenatal Care

PONE-D-24-48699R3

Dear Dr. Thai,

We’re pleased to inform you that your manuscript has been judged scientifically suitable for publication and will be formally accepted for publication once it meets all outstanding technical requirements.

Kind regards,

Ahmed Mohamed Maged, MD

Academic Editor

PLOS ONE
---

## [Editor Report · Acceptance letter]

PONE-D-24-48699R3

PLOS ONE

Dear Dr. Thai,

I'm pleased to inform you that your manuscript has been deemed suitable for publication in PLOS ONE. Congratulations! Your manuscript is now being handed over to our production team.

Kind regards,

on behalf of

Professor Ahmed Mohamed Maged

Academic Editor

PLOS ONE